# Effective Field Immobilisation and Capture of Giraffe (*Giraffa camelopardalis*)

**DOI:** 10.3390/ani12101290

**Published:** 2022-05-18

**Authors:** Francois Deacon, Willem Daffue, Pierre Nel, Ruan Higgs

**Affiliations:** 1Department Animal, Wildlife and Grassland Sciences, Faculty of Natural and Agricultural Sciences, University of the Free State, Bloemfontein SA-9300, South Africa; higgsruan30@gmail.com; 2Kroonstad Animal Hospital, Kroonstad SA-9499, South Africa; kdh@gcs.co.za; 3Free State Department of Economic Development, Small Business, Tourism and Environmental Affairs, Bloemfontein SA-9300, South Africa; nelpjvet@me.com

**Keywords:** conservation, darting, free-roaming giraffes, mortality, myopathy, veterinary procedures, risks, zoo-housed giraffes

## Abstract

**Simple Summary:**

It is known that the death rate amongst giraffes during immobilisation, capture and transportation is high. However, during this study period (2011 to 2021), 75 giraffes were captured for the collection of various samples and purposes and none of these individuals died. General experiences and lessons learned during these captures are described. Data on the knockdown times on 43 occasions of giraffe immobilisation were recorded and analysed. We hope that this shared information could help in shaping future standard operating procedures to increase the success of handling giraffes and ultimately contribute to the conservation of the species.

**Abstract:**

One of the highest occurrences of mortalities among giraffes (*Giraffa camelopardalis*) takes place during immobilisations, captures and translocations. Common mistakes, human error, unforeseen risks, the awkward anatomy and the sheer size of the animal are leading factors for giraffes’ mortalities during these operations. Many risks can be circumvented but some risks are unpreventable, often due to terrain characteristics (rivers, deep ditches, holes and rocky terrain). From 2011 to 2021, seventy-five giraffes were successfully immobilised and captured to collect biological and physiological data from eight different study areas across South Africa. A 0% mortality and injury rate was achieved and, therefore, the techniques described in this paper are testimony to the advances and improvements of capture techniques and drugs. Biological information and capture experiences were noted for 75 immobilised giraffes, of which, knockdown time data were recorded for 43 individuals. Effective and safe immobilisation requires a competent team, proper planning, skill and knowledge. In this manuscript, we address procedures, techniques, ethical compliance, welfare and safety of the study animals. General experiences and lessons learned are also shared and should benefit future captures and immobilisations by limiting the risks involved. The sharing of experiences and information could influence and improve critical assessments of different capture techniques and can likely contribute to the success rate of immobilisation and translocation success for giraffes in the future.

## 1. Introduction

One of the main goals for the continued success in the conservation of remaining giraffes (*Giraffa camelopardalis*) populations is to share the biological and physiological data acquired during field expeditions. Hands-on knowledge is often lost due to the fact that improved procedures are often not documented. One such practical experience is the capture process. Giraffe immobilisation and capture are notorious for being difficult and dangerous to the animal, as well as to the personnel, emphasised by a high animal mortality rate [1,2]. Even in more controlled ex situ environments (e.g., zoos), full immobilisation, to assist with diagnostic radiographs and hoof trims, has led to an estimated 10% mortality rate [3].

The ultimate success of the capture, transport and re-establishment of giraffes is not determined only by the capture but more often by the condition and health state of the animals before and after capture, habitat suitability at the destination, nutritional quality discrepancy between the origin and destination (especially if the nutrition at the destination is of a lower quality than at the origin), climatic unsuitability of the new environment and whether they adapt to and breed successfully in their new environment [2]. The capture success is also influenced by how they are handled, transported (if they are to be moved) and kept during and after their capture [4,5,6]. An immobilised giraffe could be exposed to myopathy (capture stress), which is mostly brought on by anaerobic respiration and the build-up of lactic acid, causing excessive exhaustion, stress and hyperthermia [7]. Myopathy could be exacerbated by a variety of underlying factors, such as disease, young or old age, advanced pregnancy, nutritional and mineral deficiencies, weaknesses caused by internal parasites and other factors that are not obvious or unknown [8,9]. These factors may have severe adverse effects on the animals, lower their resistance to stress and make them more susceptible to capture stress and exhaustion. Many animals, but especially browsers such as the greater kudu (*Tragelaphus strepsiceros*) and giraffe, lose physical condition during the dry period (winter) because of the reduced availability of food and lower energy levels in the vegetation [10]. It is therefore not advisable to capture these animals at the end of the winter when the nutritional value of the habitat is at its lowest [11] and the condition of giraffes is poor [12].

Their large size and sensitivity to drugs significantly increase the anaesthetic risk of giraffes [13]. Immobilisation-related mortality is estimated to vary between 25 to 35% [1]. Their unique anatomy creates peculiar handling problems and also puts them at risk of subluxation of cervical vertebrae during the capture operation [12,14]. The large respiratory dead space (almost 3 L in an adult giraffe) adds a physiological disadvantage to safe anaesthesia [14,15]. Another period of increased risk is the time after the immobilisation with opioid drugs. During this time, the animal becomes oblivious to obstacles and, if not managed correctly by an experienced ground support team [15], could run into dry gullies (dongas), waterbodies, over cliff edges and into fences or step into antbear holes created by aardvark (*Orycteropus afer*). Many risks cannot be managed effectively and one should then strive to minimise the risk. Participants should be aware that the operation is inherently risky.

Successful capture is therefore dependent on proper planning before undertaking an operation. Techniques for the safe and successful immobilisation, capture and translocation of giraffes have improved over the past four decades but have not proceeded without injuries and mortalities [2,14]. The immobilisation of giraffes using various drugs and cocktails (a combination of drugs) have been described [16], but controversy still exists surrounding which drugs are the best suited for the safe and effective immobilisation of this species.

One of the aims of this paper is to suggest the drug and general dosage requirements for the successful immobilisation of giraffe, which can then be marginally increased or decreased depending on the situation. A standardised operating procedure (SOP) is encouraged, in which, the dosage rates and risks are simplified and are based on the successes experienced during this and other successful projects.

The objectives of this paper are (1) to provide guidelines on successful giraffe immobilisation and capture operations, such as dosage requirements; (2) to provide adaptive measures whereby operations can be improved on; (3) to describe immobilisation and capture techniques that comply with all of the regional requirements of the animal ethical committees, animal welfare and conservation legislation and (4) to encourage a SOP, in which, the dosage rates and risks are simplified.

## 2. Materials and Methods

From 2011 to 2021, 75 giraffes were captured and immobilised at eight different study sites across South Africa. The sites, as summarised in Table 1, were mainly located within the Free State (FS) Province, with two sites in the Northern Cape (NC) Province and only one site in the North West (NW) Province of South Africa.

During the present study, all giraffes were successfully immobilised, with a 0% mortality and injury rate. Initial approaches included the use of different drugs and combinations of drugs (cocktails), such as M99 (an etorphine), Thianil (A-3080) and drug combination butorphanol-azaperone-medetomidine (BAM) (Wildlife Pharmaceuticals, White River, South Africa). As the project progressed, a preferred combination was adhered to and used in preference to the other combinations.

## 3. Results and Discussion

### 3.1. Darting and Capture

The techniques applied to immobilise giraffes vary significantly and depend on several variables, i.e., the type and duration of the procedure to be carried out and the location of the giraffes, as well as the accessibility to the giraffes. Restraint facilities are often used for giraffes in captivity (e.g., zoos) and vary from simple stalls with movable walls to highly sophisticated squeeze cages [17]. Handling immobilised animals should be under the expert care of an experienced wildlife veterinarian [18,19].

For free-ranging giraffes, two different immobilisation practices are commonly followed in the field. Both practices require the use of a remote injection syringe (dart), which can be delivered either by shooting (darting) from the ground (from a vehicle or on foot) or from the air using a helicopter or a drone [20]. Giraffes have extremely thick skin [2] and, therefore, during this study, tranquilisation drugs were delivered from a distance of no more than 40 m by means of a dart gun. The dart consisted of a 2 mL projectile syringe and a 2.5-inch 13 GA barbed side-ported needle of at least 40 mm length. When darting a giraffe, the dart should be placed in the shoulder rather than the rump, as it provides a quicker knockdown time [12].

After the chosen drug (Thianil) had effectively been released from the dart syringe, three phases of anaesthesia were recognised. The first signs (at approximately 1.5 to 4 min after darting) are what seem to be a lowering of the giraffe’s hips, hoofs trampling and high stepping gait, along with the lifting and/or shaking of the head and curling of the tail. With the use of Thianil, the average knockdown time (n = 25) was 4.02 (SE ± 1.0) minutes, as summarised in Table 2. The second phase consists of the animal becoming oblivious to environmental obstacles, such as bushes, fences, ditches, water, cliffs and the capture team’s movements/noises. Within a difficult terrain, the second phase is especially dangerous. Planning must be conducted to consider the obstacles and circumstances that may pose a danger to the animal and to be able to always keep the animal in sight. It is recommended that, where possible, capture should always be attempted away from water and on a surface as flat and even as possible; away from fences and navigable by off-road vehicles [21].

During the third phase, the animal experiences increasing levels of ataxia, slows down to a slow stumbling gait and may even fall over by itself. Whilst being in a cataleptic state, it still keeps its head balanced. With the aid of the 20 m polyester flat belt held firmly in front of the animal (i.e., tripping), the third phase is also the ideal time for the capture team to slow the animal until it falls to a prone position. Through experience gained during the various giraffe captures, the suggested length of the belt has proven efficient even when wrapped around the legs to trip the giraffe. The belt also usually sticks to the hair of the giraffe, which makes it is easy to run with and does not get tangled easily like rope.

As soon as the animal is on the ground, the antagonist to the drug must be administered to prevent hypoxia (respiratory depression) [22]. Individuals immobilised with Thianil (n = 25) were kept on the ground for an average of 11.47 (SE ± 1.0) minutes.

To add to the safety of the animals, the current study provided a minimum of two wildlife veterinarians present during each event. One veterinarian performed the darting and immobilisation from a helicopter or vehicle and the second veterinarian accompanied the ground team to the darted animal to be at hand as soon as the animal went into recumbency. The aim was to operate at a high standard of professionalism and to use skilled veterinary professionals and scientists that understood the protocols and procedures.

### 3.2. Immobilisation Drugs and Knockdown Time

Previous studies on ground-darted, free-ranging giraffes have recorded that the drug combination etorphine–azaperone had a similar induction time (first signs of drug effects and time to recumbency) to thiafentanil–medetomidine–ketamine, but was quicker than the drug combination BAM [13,23]. During the present study and summarised in Table 2, it was, however, experienced that M99 (an etorphine) had a slower induction time than Thianil (A-3080) and that the BAM (Wildlife Pharmaceuticals, White River, South Africa) combination did not ensure reliable immobilisation for free-roaming giraffes. The latter is contradictory with results recorded from previous studies [24] but could be attributed to the fact that these studies mainly focused on the immobilisation of habituated animals held in captivity under controlled environments (such as within a zoo enclosure) and not easily startled free-ranging wildlife. Wild and free-roaming giraffes require higher doses to achieve the same degree of immobilisation experienced in captive individuals [25]. We suggest that the hyaluronidase could be used in combination with M99 to speed up the induction time. Caution should then be taken to ensure that the immobilised animal is kept cool at all times, as hyaluronic acid could lead to a dramatic rise in body temperature [26]. Two suggestions for keeping the animal cool include providing shade with a mobile awning (or umbrella) or rubbing 20 to 40 L of water onto the hair, or both [27].

Based on user preference and experience, a great variety of drugs, drug combinations and dosages are used for the immobilisation of giraffe. Opioid drugs are usually the main component for the immobilisation in the field. Giraffes are very sensitive to opioid drugs and its dosages can cause profound respiratory depression [15]. Giraffes also have limited exercise tolerance [15], which necessitates the need for higher doses for quicker induction times. For this reason, the opioid drugs used for field immobilisation are usually reversed by administering an opioid antagonist drug intra-venously as soon as possible after recumbency [15,28]. Hypoxia quickly sets in if the effects of opioid drugs are not reversed, and it is therefore advised that the antagonist should be administered within 5 min of opioid exposure, even if only partially reversing the effects [28]. Opioid antagonists can either be pure or partial antagonists. Pure opioid antagonists reverse all of the opioid receptor effects and result in an animal that becomes fully awake. Partial opioid antagonists only antagonise certain specific receptor types, whilst leaving other receptor types unaffected. A trade-off therefore exists between higher dosages with quicker induction times but lower muscular fatigue, as well as more severe respiratory depression, and lower dosages with less respiratory depression but longer induction times and possible exertion myopathy complications.

In Southern Africa, common practice for drugs and dosages for the chemical immobilisation of healthy, wild giraffes includes 8–12 mg of etorphine hydrochloride (M99/Captivo) [29]; 60–80 mg of the tranquiliser azaperone [15] (although the adding of any sedatives or tranquillisers to the dart mixture has mainly fallen to disuse), with a 16–24 mg dose of diprenorphine (M5050/Activon) as a partial opiate antagonist. The use of 12 mg of A3080 and 100 mg azaperone as an immobilising drug and tranquiliser combination has been recommended as well [30], but the addition of azaperone is seldom used any more. An alternative combination is the use of 5 mg Thianil + 5 mg medetomidine and 4 mg Thianil + 4 mg medetomidine for adult giraffe bulls and cows, respectively. This combination has a slower knock-down effect but is reliable and provides a much more cost-effective alternative to the use of the high dose (12 mg) of A3080. The highly specific receptor atipamezole or the less specific yohimbine (α-2-adrenoceptor antagonists) are used to reverse the actions of this drug combination [31].

During this study, initial approaches included experimenting with different drugs and combinations of drugs (cocktails), starting with only etorphine hydrochloride, combining etorphine hydrochloride and thiafentanil (Thianil; Wildlife Pharmaceuticals) and combining Thianil and azaperone. During 25 immobilisation events where Thianil was utilised on its own, it was shown that immobilised giraffes could safely spend anywhere from less than a minute (removing collars) to more than 20 min on the ground, and hence became the drug of choice. Fourteen mg and eighteen mg Thianil were administered for females and males, respectively. Thiafentanil (A3080/Thianil), is a synthetic opioid similar to etorphine in both potency and other properties [30]. The animal’s response to this drug is considered to be more predictable and it has an induction time that is significantly shorter than that of other opioids by as much as 50% [32]. The reasoning for the high doses and use of Thianil alone is the predictability of what to expect and to be able to respond more easily with the reversal. When different mixtures (cocktails) are used, the predictability of the drug combination is complicated and increases the risk of not being able to be reversed. When using a high dosage, as described above, delays in providing the antagonist (>90 s) should be minimised. During the present study, the ground veterinarian immediately injected 100 mg naltrexone (Wildlife Pharmaceuticals, White River, South Africa) directly into the jugular vein and 50 mg into the triceps muscle.

Limitations exist on the physical aspects of using dart guns to administer the immobilising drugs. The use of a heavy dart is usually the limiting factor on gaining enough distance and penetration effect. Preferably, trying to shoot and administer the drug to a giraffe further than 40 m away should be avoided. Experience in handling a dart gun is needed to ensure that the dart reaches its target correctly the first time. With a missed shot, it becomes more difficult to approach the individual or herd again, as they usually start running away. A good shot placement, such as in the shoulder muscles [22] or near the base of the tail, will lead to quicker absorption into the blood stream and, ultimately, a quicker knockdown time. An increased darting distance increases the risk of poor shot placement. The reliability and accuracy of the type and brand of the dart should also be taken into consideration. Dart malfunction may be a result of various factors, such as the incorrect loading of the dart, a manufacturing error or a skin plug blocking a needle. During the present study, the 2cc 13-gauge type P (pneumatic) and type C (charge) darts with dual ports and a needle length of 2.5″ were sufficient. Side-port needles should be utilised in order to avoid the formation of skin plugs.

A misfire and the inaccurate calculation of the dose needed (especially under-dosing), as well as poor dart placement can increase the risk of an operation by delaying the time to immobilisation and leading to excessive running. The above may cause the animal to exhaust itself by running too far, or it could overheat before the drug becomes effective. Both consequences could lead to the death of the animal from exhaustion or capture myopathy. If the first attempt by the ground vehicle is ineffective, a helicopter on standby is suggested to administer the second dart. Although expensive, it is encouraged that, when a giraffe cannot be darted from the ground or from a vehicle, a helicopter should be used. The tracking of immobilised animals in dense bush or rocky areas may damage the capture vehicles and put the animal itself in danger. In a worst-case scenario, it could also result in the antagonist not reaching the animal in time to reverse the severe respiratory depressant.

### 3.3. Monitoring of Vital Signs

Breathing is considered as the most important vital sign to monitor whilst a giraffe is immobilised [33]. Of all of the vital signs, breathing is the first to be affected negatively, after which, cardiac function and blood pressure will follow. Oxygenation is influenced by the depth and rate of breathing. If breathing is deep and regular, it is unlikely that there will be other serious physiological abnormalities. Ideally, the animal should be using maximum lung capacity with deep, long breaths. Shallow breathing results in a dead space of air mostly moving up and down the trachea, without reaching the lungs efficiently [19,33].

Members of the capture team restrained each animal in such a way that its lungs could expand as much as possible. The breathing was monitored most easily with the giraffe’s head on one team member’s lap or mobile stretcher, with one hand close to the nostrils, without obstructing the nostrils. This monitoring assisted in feeling the movement of hot air and counting the respiratory rate. Knowledge of the respiratory rate and monitoring it throughout the restraining period are of vital importance (should be around 12 to 20 deep breaths per minute), as it will be affected if the giraffe is ill. The head should be kept stable, with the other hand having a firm grip over the ossicones. Overcrowding around the head and nasal area should also be avoided. In hot humid conditions, the use of a battery-operated leaf blower is highly advantageous to cool a hyperthermic animal down. The placement of the earplugs should prevent the animal from being startled by these external sounds.

Every few minutes, the normal body temperature was measured by placing a digital thermometer deep within the rectum. For an adult giraffe, the normal body temperature should be between 37 to 38.6 °C, not exceeding 39 °C [34]. Normally, if an animal reached 38.6 °C, it was quickly lowered by pouring copious amounts of cold water on the body and a controlled amount on the head and ensuring adequate air flow over the wet areas. We considered any temperature 39 °C or above as critical and would not allow the animal to remain immobilised if the temperature stayed above that for more than 2 min. However, when running for long periods of time during the darting process, most individuals would typically have temperatures of around 40 °C. By using the techniques described above, the core temperature should stabilise below 39 °C within 2 min. If not, we recommend abandoning the capture and releasing the animal.

Unlike horses’ gums, which have a pink colour, the normal colour of a giraffe’s pigmented gum is a dark blue/purple/greyish colour. These darkish colours limit the value of using the oral mucous membranes for the evaluation of blood oxygenation, peripheral circulation or cardiovascular function. White, dark red, bright red or too dark colour variations indicate various forms of shock that the animal could be experiencing. If the mucous membranes in a giraffe’s mouth turn light grey/whitish in colour, immediate actions should be taken to improve ventilation [19]. A capillary refill test by pressing and releasing with a finger every minute or two on the gums and confirming if the colour changes back to normal is a vital part of the monitoring. The colour of the gums should return back to the original colour within two seconds after release. The mucous membranes on the inside of the giraffe’s nostrils could also be monitored, as these remain pink when blood oxygenation is sufficient and should not be purple or darkening at all. The reaction and response from the animal (eye reflexes) were tested by gently touching the canthus of their eye with one finger. If the giraffe does not respond to the touch, then the animal could be suffering from possible hypoxia, or too deep anaesthesia, and should immediately be released if still possible.

The suggested body positioning, which includes lateral recumbency and limbs being removed underneath the animal, complicates the monitoring of the heart rate on both sides. The heart rate was monitored with a stethoscope by placing it on the left side of the abdomen, just behind the elbow (if the animal was not lying on the left side). To ensure that appropriate blood circulation occurs, the heart rate should be steady and regular and, on average, should be ranging from 60 to 120 beats per minute [35,36]. Importantly the heart rate can vary amongst individuals, as it will be determined by the distance and terrain that the animal had run before and during the immobilisation process. Higher heart rates are often a physiological response to hypoxia, excessive running, stress or low blood pressure. In the absence of a stethoscope, the pulse rate was taken from the lingual artery, located on the bottom side of the jaw where it crosses over the bone. Although it is considered a very crude field method, adequate blood pressure could also be determined by means of visual observations on the prominence of arteries and veins around the head and nasal area.

### 3.4. Body Positioning and Restraint during Recumbency

When darted on the ground, the animal is positioned on its side and the neck and head of the animal are secured. Eyes are covered and plugs are inserted into the ears to limit stimulation from external sources, which cause the now fully conscious (after intra-venous administration of the antagonist drug) giraffe to struggle and kick. To limit the giraffe’s exposure to startling sounds, cotton wool, placed within a soft sock and rope (to pull out quickly) or cloth against the ear, were used.

The giraffe was kept down by placing weight on the neck and body (carefully and not putting excessive pressure on the trachea and rib cage, which would interfere with efficient ventilation), whilst still keeping its head and neck elevated. Immobilised ruminants, especially if food and water were not withheld before anaesthesia, should be kept in sternal recumbency, i.e., their heads and necks must be held higher than their chest to avoid bloat and regurgitation [22]. For giraffes, this positioning is, however, not possible, as they are usually fully or partially revived by administering opioid antagonists soon after recumbency [15]. After letting the animal fall to a prone position, all limbs should be removed from underneath the animal as soon as possible to limit the pressure on its blood circulation. Numb limbs should be avoided, as it will negatively impact the ability of the animal to stand up again. By making use of a mobile stretcher and with the assistance of a few people, the head and upper neck should be elevated to prevent passive regurgitation [2]. Under normal circumstances, even when sleeping, giraffes do not lay down with their heads flat on the ground [37]. By elevating the head and neck slightly, it therefore may aid in blood pressure management, as well as assisting with the veterinary monitoring of the animal.

Constant attempts by the animal to try and stand upright, kicking and struggling movements indicated signs of discomfort and stress. These were addressed immediately by making the animal as comfortable as possible (removing twigs, thorns, rocks, etc.), making sure the eyes were covered properly, the ears were plugged and the team were being as silent as possible.

With the completion of the data collection checklist, everyone swiftly moved away from the animal, removing the blindfold and ear plugs and ensuring that the dart has been removed as well. One should keep in mind that the giraffe would be completely conscious at this time and that the animal would then stand up. In cases where it has difficulty getting into sternal recumbency, aid must be provided by the capture team members.

### 3.5. Transport

The immobilisation agent that was used should be fully reversed before attempting to guide and load the animal through the use of ropes [13]. The administration of neuroleptics should be avoided during transport due to the risk of disorientation and collapse as a result of unsteadiness. Calmer individuals (restrained giraffe) benefit the safety of the animal, as well as the capture team during loading and transport operations [13]. When multiple animals are transported in the same transport crate, it is crucial that the characteristics of the drug used to calm them and their present state of awareness are accounted for. Individuals under the influence of long-lasting drugs should not be kept or transported along with individuals that were administered short-acting drugs, and even more so for individuals of which the effect of the drug has already worn off [4]. A short-acting tranquiliser for the transportation of giraffes that has been captured recently can be provided, but the most recent tendency is to translocate giraffes without sedation because a sedated giraffe falling down in a crate with other giraffes could result in injury or death. Although sedation should assist in relieving the individuals from stress and reducing aggression, [2] the risk should be weighed against the benefit. Giraffes transported in groups are much more relaxed than those transported in single crates. Sedation is seldom necessary in grouped giraffes during transport. Regular stops during transit are recommended, as it will provide opportunities to detect unforeseen circumstances and then take the needed action [38].

## 4. Conclusions

Proper planning before undertaking the capture and immobilisation of giraffes is crucial. Their large size, peculiar anatomy and their sensitivity to drugs significantly increase the anesthesia risk of giraffes with a known immobilisation-related mortality rate of an estimated 25 to 35%. The techniques described in this paper provide insight on the advances and improvements of capture techniques and drugs. The sharing of experiences and information will hopefully be of assistance to influence and improve critical assessments of different capture techniques by positively contributing to the success rate for future captures, as well as giraffe handling, training, management and conservation efforts, within in situ and ex situ environments. Recommendations for a SOP for giraffe immobilisation is summarised and shared as part of the Appendix A.

## Figures and Tables

**Table 1 animals-12-01290-t001:** The sites, located in the Free State (FS), North West (NW) and Northern Cape (NC), at which, giraffes (n) were captured and immobilised from 2011 to 2021.

Site Name	Province	Males (n)	Females (n)	Year of Immobilisation
Woodland Hills Wildlife Estate	FS	2	0	2011; 2012
Khamab Kalahari Reserve	NW	0	16	2013; 2014
Willem Pretorius Nature Reserve	FS	0	4	2014
Sandveld Nature Reserve	FS	0	2	2015
Amanzi Nature Reserve	FS	1	2	2015; 2018
Doornkloof Nature Reserve	NC	0	1	2017
Theunissen Mpogo Nature Reserve	FS	1	0	2017
Rooipoort Nature Reserve	NC	14	32	2017; 2018; 2021
Total (75)		18	57	2011 to 2021

**Table 2 animals-12-01290-t002:** Recorded data from drugs and drug cocktails used in the field during the present study on 43 occasions of giraffe immobilisation.

Drug/Cocktail	Antagonist	Number of Giraffes (n)	Knockdown Time Average	Knockdown Time Range	Time on Ground (Average) (Hours:Minutes:Seconds)
Thianil, Etorphine, Azaparone	Naltrexone	1	00:04:36	00:04:36	00:07:01
M99, hyaluronidase, Azaparone	Naltrexone	9	00:04:41	03:28–06:56	00:14:23
M99, hyaluronidase	Naltrexone	7	00:14:16	04:55–21:30	00:19:25
butorphanol-azaperone-medetomidine	Naltrexone	1	00:50:30	00:50:30	00:09:30
Thianil	Naltrexone	25	00:04:02	02:50–06:00	00:11:47
Total		43 *			

* During the study period, 75 giraffes were immobilised and handled by the research team for various scientific procedures. From the 75 giraffes, knockdown time data were recorded and analysed for 43 giraffes.

## Data Availability

Not applicable.

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
