# Peer review of "Effective Field Immobilisation and Capture of Giraffe (Giraffa camelopardalis)"

_animals, 2022, doi:10.3390/ani12101290_

Round 1
Reviewer 1 Report
Dear authors, all the comments are provided in the word document that is here attached. If you will be able to provide photos and videos as Supplmentary material, please be aware of copyright rules and obtain from all the people in the photos an informed consent.
Best, the reviewer

Reviewer 2 Report
The manuscript as written requires two major changes: 1) data needs to be added and summarized, 2) removal of repetition. As written, the manuscript is a useful contribution but with limited scope. With the addition of core findings (i.e., data with means, SE, ranges), the experience of the authors would be magnified many times over. There is a significant amount of practical and useful information in the manuscript and with some careful editing, additional clarity on key points, statistics (even the most basic), a table or two, then, the manuscript will make a significant contribution to the handling of giraffe. I commend the authors for summarizing their experience and urge them to undertake a revision that will benefit conservation and animal welfare even further.
Abstract
19 - I would prefer to see the binomial name noted
27-28 – I think competent approximately equals experienced but if you wish both, I think “competent and experienced” – I would only use “competent”.
29 – insert “study” before animal (but I think it should be “animals”).
39-40 “This knowledge is often lost due to the fact that it is often not documented.” – The flow / logic here is unclear. It sounds like people are catching giraffes and not publishing. Is this the idea?
44 – “ but more often by the condition” - the source of this statement
43-57 – none of this perspective is supported by the literature. Please provide references. This level of insight cannot be attributed to common knowledge. (e.g., there are many papers on capture myopathy that could be cited).
52 – please be clear when starting sentences with “This” – specify the point clearly.
57-8 – “greater kudu” – binomial name needed.
64 – “Their unique anatomy creates unique…” – drop 1 use of unique – it’s a bit tautological.
64 – delete “particular”
69-70 – “dongas” – not a universally know term (I had to look it up) – explain to make it clearer to a general audience. Keep in mind, people catching other large mammals may benefit from your experience with giraffes.
70 – here is where binomial names are needed – “antbear” doesn't mean anything to me – aardvark does but Orycteropus afer is unequivocal.
71-2 “Role players should be aware that the operation is inherently risky.” – reword – this sounds like something from a video game as written. “Participants” – not “role players”.
77 – delete “a lot of”
82 – delete “at hand.” – adds nothing.
96-8 “Giraffe is an introduced species in the Free State and North West Provinces [10, 11] but evidence suggests that giraffe had, however, occurred naturally westwards of the Free State, in the Northern 98 Cape Province [12].” - the relevance of this is unclear. Why does it matter to the study? I suggest deleting to retain focus on the core issues.
103 – “A total of 75 giraffes” – we know this from Table 1 and line 93 – remove repetition. We also know it’s 10 year period.
105-6 – I would think the drugs and combinations used would be listed in the methods not only the results.
108 - 16.1 (+/- 1.0) – what is the plus / minus – SD or SE ? To me, however, the duration of capture is a result not a method.
110-112 “Biological and physiological samples collected during the immobilisation of these animals greatly contributed to the overall understanding of these unique mammals and will be discussed separately.” Delete – this is irrelevant to the study objectives and it’s not a method as presented.
113-5 – “This article provides some insight on the successful handling of giraffe when captured, immobilised and released.” Delete. This is not a method but objectives and we know this already (i.e., repetition).
129 – “this study” – do you mean your study or the ones cited. Unclear.
137 – “It is argued” – perhaps you mean “We suggest…” – it is unclear who is arguing the point as written. Is it the previous reference or the authors of the present manuscript.
139 – “kept cool” – do the authors have any recommendations? This is a major issue to gloss over.
141 – delete “always be able to” – they just should do it.
141 – technically, monitoring does not have to be done by a veterinarian – just individuals with appropriate training (e.g., veterinarian technician, skill professionals).
146 – “or a drone.” – a reference for drone darting, which is new, is required.
147 – “animal should then be casted” – reword / explain – I’ve no idea what this means.
161-2 “The aim was to operate on a high standard of professionalism and to use 161 skilled scientists that understood the protocols and procedures.” I do not usually consider veterinarian as “scientist”. They are animal medicine practitioners. Similarly, medical doctors are not scientists even though they may have science-based training. Reword.
176-7 – delete “e.g., bushy or 176 rocky areas, close proximity to fences and water bodies.” this point is made several times.
177-8 – Shorten: “Planning ahead and taking into 177 account all obstacles and circumstances that may pose a danger to the animal, must receive special attention.”
182 – “a seatbelt” – this doesn’t make sense. Explain what you mean by a “long belt” – how long? 1 m or 20 m and how wide 2 cm or 10 cm or ? What material? Inadequate detail.
186 – delete “to set in”
217 – “gain its feet,” to “stand”
221-2 “By following this approach, even a small but experienced team, should have great success with transporting giraffe via a truck or transport crate.” Delete – this is unfounded speculation. 1) “experienced” is undefined and subjective, 2) “great success” – meaning is unclear. The issue of “success” is based on the past experience of handling by the authors. That is sufficient context. The manuscript is overly wordy and would benefit from shortening to the core points and removal of redundancy would improve the utility of the work.
232 – “could be disastrous.” reword – remove the drama. What do you mean here? Injury or mortality? If so, use such terms.
232 – “Once the first animal falls down, others may trip over the one already down, leading to an entangled mass of giraffes unable to gain their feet and being trampled by those still on their feet.” Shorten or remove. To me, this sounds like speculation. The point is injury or mortality.
238 – 9 “Regular stops during transit are recommended, as it is essential to provide timely actions to prevent further disaster, if something goes wrong” What disaster? Further, why stop? Don’t you mean periodic checks on animal welfare? Stopping does nothing in and of itself – it’s the checking on animal status. Reword.
242-5 “ With the collection of samples and fitting of collars, members of the capture team, such as the collar technicians, veterinarians and people assisting with the restraining and covering of the eyes and ears, were involved.” Poorly worded.
247-9 “Its head was kept from the ground by means of manual lifting or the use of a stretcher. Once on the ground, the antagonist drug was immediately ad-248 ministered intra-venously.” These points were already made (e.g., using the stretcher (lines 203 and 273)) – remove redundancy.
255-6 “various scientists from different disciplines collected their data.” wordy and uninformative. Collect samples – end of issue.
269-70 – “The routine administration of oxygen is always beneficial to anaesthesised giraffe.” Fine. How? What flow rate?
277-280 – Specifics are needed. At what temperature is an animal considered hyperthermic? Is this a rectal temperature? Further, leaf blowers are incredibly noisy so this is not an issue? Clearly, cooling the animal is the priority but at what point do you abandon the capture and release the animal? If animal welfare is the key objective, release is likely a better action than leaf blowers and continued restraint. Logic on this point is unclear / poorly articulated.
293 – “Eyes should be bright and clear from any muck or mucus.” – how is this related to the capture and what would you do if there is mucus in the eyes? If there is detritus from capture, what action? Eye wash?
301-3 – “Stomach sounds and growling are normal and indicative of a healthy gut. If these sounds become fainter or non-existent, it might also be indicative of illness.” Illness is not the topic of this manuscript.
312-4 “Like all mammals, the pulse can be taken for 15 seconds and then multiplied by four to calculate the heart rate in beats per minute.” This is trivial. It could be taken for 20 seconds and multiplied by 3. Further, I would argue 15 second is too short a time for accurate measurement. Avoid trivial statements such as these. Brevity would be welcome.
314 – “Visual observations were also made to confirm adequate blood pressure by the prominence of arteries and veins around the head and nasal area (although very crude, it is a useful field parameter to keep in mind).” I’ve no idea what this means.
341-2 “It is against this background, that the science of immobilisation becomes an art [7].” Delete even if a reference says such a thing. Nonsense.
343-55 – no recommendations on dose by age, sex or reproductive status? This is quite odd and makes the work less useful. A table of doses, induction times, body temperatures relative to ambient, mean peak temperatures (SE and range), mean times to reversal and other such relevant data are a mandatory addition to the work. There is no data presented that allow others to gain insights. As written, the work is limited in utility by the generic nature of the findings. With 75 animals successfully handled, present the data on capture as a guide. Without this information, the manuscript is has limited utility beyond generalizations.
356 – “ During this study, 75 giraffes were successfully immobilised, with 0% mortality rate.” Yes, stated MANY times. Remove redundancy. Cut manuscript length by 10-20%.
369-74 – Condense – this is stated many many times. “Great planning should, however, be used when deciding on using a high dosage as described above. Any delay in being able to provide the antagonist (> 90 seconds) could result in death of the animal. Care should be taken to ensure that the tranquilized animals can be reached within a short time span and should not be compromised by an animal which had run into a terrain (e.g., 373 bush or gully) which cannot be quickly reached.”
275 “ stand-out” to “major”
378 - “the vein” – which vein?
381 – yes “(0% mortality rate),” is stated MANY times – it is beyond tedious at this point.
382–5 “there are many advantages of using chemical immobilisation. Disadvantages, however, do exist as well. A major disadvantage for the use of chemical immobilisation is the potency of the drugs and risk to humans, cost and strict control.” Condense or delete. Either an animal needs to be caught and handled or it doesn’t. These issues are generic.
388-91 – condense – not overly relevant “Current legislation dictates that these immobilising drugs are classified as Schedule 6 medicines and ensures that it may only be sold to and used by a registered veterinarian. A high level of professional expertise and knowledge is needed in the handling of the darting equipment, immobilising drugs and the immobilised animals after capture.”
395-405 – reduce by ½ - be clear and directed. Yes, avoiding poor dart placement is an issue but what is good dart placement?
406-13 – because the manuscript provides no details on the 75 captures, the reader has no idea about the study captures (e.g., how many required 2 doses, how many were caught from the ground, how many by helicopter). Provide the key results. As written the manuscript is more of a thought experiment than a work of science with data. I see general utility in the manuscript but it would be far more useful with actual information.
415-7 – “To avoid the dart making a skin plug, it is suggested that the tip of the dart needle 415 should be bent slightly which will prevent a piece of skin entering and blocking the needle.” Use a side-port needle. This is nonsensical – any skin will be blown out by the drug entering the needle.
417 – line 167-8 already makes this point and states the shoulder is the preferred dart location – again, repetition is unwelcome.
419-20 “The successful penetration of major blood vessel, e.g., near the base of the tail, will also reduce the knockdown time.” This contradicts aiming for a shoulder. Clarify.
Conclusions – this section adds little (except additional repetition). Cut by ½ or remove.
Reviewer 3 Report
Dear authors,
Thank you for taking the time to compile this paper. I believe it will of great benefit to the wildlife community and strongly advocate the sharing of expertise in this manner.
Main points:
For this paper to be of use to other practitioners more details are needed on the methods/protocols you used - how did they change? when did they change? what drugs/cocktails of drugs did you use? why? did this differ in different contexts?
The results/discussion are then quite long and a bit repetitive in places. I suggest this needs some refinement. Transport conditions would make more sense at the end, rather than in the middle. Monitoring of vital signs should be incorporated into the 'immobilisation and monitoring' section. Chemical immobilisation also needs to come with the immobilisation section. I am unclear on why the section on sample collections is needed, if the point of this paper is to provide a protocol I think this section needs to more clearly evaluate the successful/unsuccessful protocols and define why you are recommending the points for best practice that you are. There is repeated reiteration of the need for the reversal drug - that just needs to be stated once (in an appropriate place)
Citations are needed throughout the introduction and results/discussion sections (e.g. L45 - 57, section 3.7< L298)
Line-by-line comments:
L124: under what conditions? wild? what sort of variables?
L171: what was the effect of the dosage/combination?
L186: from setting in
L206: how/why?
L217: sentence has been said previously, not needed here
L293: you have said previously the eyes should be covered
L360/361: why?
L385: unsure why strict control of drugs is an issue, when this process should only be being undertaken by licenced practitioners?
Round 2
Reviewer 3 Report
Dear authors,
Thank you for taking the time to make modifications to your manuscript and respond to my comments. I think the manuscript is generally much clearer but there are still some areas where further clarity is needed, and there is still some repetition of the process which could do with being refined further.
L28: I am unclear why you are referring to the 75 giraffe at all, why not just focus on the 43?
L109: suggest remove reference to Table 2 - but make reference to this in the paragraph which relates to drugs used in immobilisation
Section 3.1 – I think this needs separating as at the moment it is very long. I would have ‘monitoring of vital signs’ as a separate section. Your recommendations arising from your work in relation to immobilisation then need to be more clearly stated, as at the moment they are lost
L132/133: citation needed
L195: initial approaches – is that your study or other work? need to clarify
Table 2: column on deaths can be deleted as you have already said in the MS there were none. Times – are they minutes or seconds?
L228: of various factors
Paragraph starting L245 – citations needed
L253 – easiest is the wrong word here – maybe ‘most easily’?
L280/281: citation needed
3.2 – darting and initial drug effect section doesn’t need to be in its own section – this can go in the immobilisation, but be mindful of not repeating
L306: skin
L315 – why 25 animals?
L327: restraint
L350 – 352: information not relevant to the section
L363 – 365: first two sentences are not needed. Suggest moving sentence ‘after which’ to after the sentence on immobilisation
Conclusion needs a summary of your actual recommendations, at the moment the message is slightly lost. In the aims of your work you mentioned development of a SOP, but this seems to be lacking. Being clearer about your recommendations would help to move towards this point, although I appreciate you cannot develop SOPs from this piece of work alone, having the general ideas condensed and easily accessible will be useful to other practitioners.
